# Adversarial Inverse Reward-Constraint Learning with Reward-Feasibility Contrast Prior Inspired by Animal Behaviour

## Abstract

The behaviour of natural and artificial agents is shaped by underlying reward systems, which signal rewards based on internal and external factors, driving reward-oriented actions. However, real-world scenarios often impose constraints that reward alone cannot capture. While existing inverse (constrained) reinforcement learning methods can recover either rewards or constraints from demonstrations, the simultaneous inference of both remains unexplored due to the complexity of inference and the lack of knowledge of their relationship. To address this gap, we propose a novel algorithm that simultaneously infers both rewards and constraints within an adversarial learning framework, where both are updated through a policy optimisation process guided by expert demonstrations. Crucial to this framework is the introduction of the "reward-feasibility contrast prior," a hypothesis that correlates rewards and constraints. It is inspired by patterns observed in animal behaviour (particularly meerkats), positing that states with high rewards nearby are more likely to be associated with weaker feasibility (stronger constraints). Our experiments on virtual robot control tasks with safety constraints and real-world animal behaviour data with spatio-temporal causal constraints validate our proposed framework's effectiveness and the reward-feasibility contrast prior hypothesis. The results show accurate recovery of rewards and constraints, reflected by strong alignment with expert demonstrations and a low rate of constraint violations. Additionally, the performance improvement by embedding this prior into other inverse constraint inference methods further confirms its general effectiveness.

## 1 Introduction

Understanding the motivations behind the behaviour of agents, particularly natural beings, and designing intelligent agents that behave as desired are key desiderata in artificial intelligence, robotics, neuroscience (Maia & Frank, 2011) and ethology (Mori et al., 2022). Inverse reinforcement learning (IRL) (Ng & Russell, 2000) offers a framework for achieving this by recovering the underlying reward function – often considered the most succinct representation of the objective of the behaviour (Abbeel & Ng, 2004; Fu et al., 2018) – from demonstrations, making it a powerful tool for interpreting behaviour and generating synthetic actions in real-world scenarios, such as robot control (Bogert & Doshi, 2014; Kretzschmar et al., 2016) and animal behaviour modelling (Inga et al., 2017; Ashwood et al., 2022). However, real-world behaviour is often constrained by factors beyond rewards, such as avoiding hazards in robot navigation (Bogert & Doshi, 2018) or environmental and cognitive limitations in humans and animals (Niv, 2009). These constraints cannot be fully captured by rewards alone and must be considered alongside rewards to fully capture behaviour dynamics.

Inverse constrained reinforcement learning (ICRL) (Scobee & Sastry, 2019; Malik et al., 2021) offers a promising approach for inferring constraints from demonstrations. However, most existing ICRL methods rely on the assumption of known reward functions (Malik et al., 2021; Liu & Zhu, 2022), whereas real-world scenarios, such as animal behaviour modelling, often lack such reward signals. While few approaches assume unknown rewards (Xu & Liu, 2024; Lindner et al., 2024), they impose linear assumptions on the environment dynamics and reward functions, reducing the problem to a linear programming-like formulation and involving convex optimisation, which significantly limits their expressive power and applicability in real-world settings. As IRL and ICRL remain

Figure 1: The proposed adversarial inverse reward-constraint learning framework simultaneously updates reward and constraint functions through policy optimisation to reproduce demonstrations. The reward-feasibility contrast prior, inspired by animal behaviour, emphasises that high rewards often correlate with high constraints, shaping the correlation between them.

complementary: each inferring either complicated rewards or constraints but not both simultaneously, which underscores the need for a unified approach.

Unifying IRL and ICRL presents two primary challenges. First, an efficient framework is needed to infer both complex rewards and constraints simultaneously. While IRL and ICRL have practical implementations individually (Fu et al., 2018; Malik et al., 2021), their mechanisms are not inherently designed to handle unknown constraints or rewards, preventing their combined application. Second, without prior knowledge of the correlations between constraints and rewards, there is a significant risk of deriving counterfactual solutions that diverge from reality. For example, in behaviour modelling, multiple reward-constraint combinations might explain the observed demonstrations, but not all align with the true underlying dynamics.

To address the first challenge, we extend the adversarial IRL framework (Fu et al., 2018) to infer both rewards and unknown constraints simultaneously. The adversary updates these components based on agent-generated samples, while the agent's policy is optimised using these updates, creating a more integrated inference process. To tackle the second challenge, we investigate the correlations between rewards and constraints in natural settings, focusing on meerkats due to their complex social structures (Drewe et al., 2009; Madden et al., 2009; 2011). We extract spatiotemporal transitions from footage data of a meerkat mob (Rogers et al., 2023), and our analysis reveals that long-distance movements, despite being less feasible, are likely driven by high rewards. This observation forms the basis of our hypothesis, the *"reward-feasibility contrast prior."* This prior, independent of our proposed framework, reflects the common understanding that higher rewards are often tied to greater risks (Lopes, 1987). We formalise this prior as a regularisation term and integrate it into our framework.

Our primary contribution is a novel algorithm that unifies IRL and ICRL, which we name *Adversarial Inverse Reward-Constraint Learning* (AIRCL) as illustrated in Figure 1. Unlike prior methods in IRL and ICRL, AIRCL allows for the efficient and simultaneous inference of complex unknown rewards and constraints (Section 5), while capturing their correlations through the reward-feasibility contrast prior (Section 6). We evaluate AIRCL on simulated benchmark robot-control tasks with safety constraints and real-world animal behaviour modelling tasks with spatio-temporal causal constraints (Section 7). Our method effectively recovers reward functions and reveals underlying constraints, evidenced by reproducing behaviours that closely match the demonstrated behaviours while adhering to the inferred constraints. Experimental results confirm its effectiveness in modelling natural and robot behaviours and its broad effectiveness. In addition, the performance improvement of ICRL by embedding the proposed prior further confirms its versatility and general effectiveness.

## 2 RELATED WORK

Our work is grounded in inverse reinforcement learning (IRL) (Ng et al., 1999), which originates from inverse optimal control (Kalman, 1964), where the goal is to identify the function optimised by a given system governed by some known control law. In machine learning, IRL was formally introduced by Ng & Russell (2000) and applied to imitation learning (Ho & Ermon, 2016), also known as apprenticeship learning (Abbeel & Ng, 2004), to train agents that perform as well as human experts. Compared to traditional imitation learning methods like behaviour cloning (Bain & Sammut, 1995) and Dataset Aggregation (DAgger) (Ross et al., 2011), which directly mimic behaviour, IRL

offers better explainability and robustness, as the recovered reward function not only interprets agents' motivations but also remains invariant to changes in environmental dynamics (Fu et al., 2018). Early IRL approaches relied on margin optimisation and assumed linear reward functions (Syed & Schapire, 2007; Syed et al., 2008; Ratliff et al., 2006), though these were ill-defined due to the potential for multiple reward functions explaining the same demonstrations. Maximum (causal) entropy (MaxEnt) IRL (Ziebart et al., 2008; 2010) addressed this issue by selecting the reward function that maximises the entropy of the distribution of demonstrations. Variants of MaxEnt IRL have been developed for scenarios such as non-optimal demonstrations (Boularias et al., 2011), multi-objective settings (Babes et al., 2011), and specific Markov decision processes (Dvijotham & Todorov, 2010), as well as Bayesian (Ramachandran & Amir, 2007) and Gaussian process-based methods (Jin et al., 2015). However, MaxEnt IRL is computationally challenging for large or continuous state spaces due to its dependence on forward RL subroutines. Deep learning advancements, such as guided cost learning (Finn et al., 2016b) and adversarial IRL (AIRL) (Finn et al., 2016a; Fu et al., 2018), resolved this limitation through sampling-based approximations to MaxEnt IRL, which we also leverage in our approach. Extensions to meta-learning (Yu et al., 2019b), multi-agent (Yu et al., 2019a), and many-agent settings (Chen et al., 2023; 2024) have followed. However, these methods focus exclusively on rewards, overlooking constraints. Our approach builds on this foundation, extending IRL to infer both rewards and constraints, offering enhanced explainability and practical utility.

This paper is closely related to inverse constrained reinforcement learning (ICRL) (Liu et al., 2024), which focuses on recovering unknown constraints from demonstrations. ICRL is the inverse problem of constrained reinforcement learning (CRL) (Tessler et al., 2019), where the goal is to optimise rewards while satisfying constraints related to safety (García & Fernández, 2015), security (Lei et al., 2023; Zhang et al., 2023), fairness (Jabbari et al., 2017), or other considerations (Qin et al., 2021). The ICRL problem was studied by Scobee & Sastry (2019) in discrete settings, with the term ICRL introduced by Malik et al. (2021) for continuous state-action spaces. These and other works (McPherson et al., 2021; Baert et al., 2023) use the MaxEnt IRL framework to identify constrained state-action pairs. In addition to MaxEnt IRL, alternative methods include linear programming (Lindner et al., 2024), robust optimisation (Xu & Liu, 2024), Bayesian methods (Papadimitriou et al., 2022), and generative models (Xu & Liu, 2023), with some ICRL extensions addressing multi-agent scenarios (Xu & Liu, 2024; Liu & Zhu, 2024). However, many ICRL approaches assume known or linear reward functions, limiting their practical use in real-world applications. Our work overcomes this limitation by enabling simultaneous inference of both rewards and constraints without these restrictive assumptions.

Our research is also connected to the literature on human and animal behaviour modelling using reinforcement learning, both in physical (Ashwood et al., 2020; Mori et al., 2022) and psychological contexts (Niv, 2009; Maia & Frank, 2011; Gershman & Daw, 2017). Unlike previous studies focusing on directly building reinforcement learning methods to reproduce human or animal-like behaviour, our approach seeks to uncover intuitive clues about the correlation between rewards and constraints from observed meerkat behaviour in their habitat. Specifically, from the phenomenon of relatively frequent long-distance movements, we formalise the reward-feasibility contrast prior, which serves as a critical regularisation component in our proposed framework.

## 3 PRELIMINARIES

### 3.1 INVERSE REINFORCEMENT LEARNING

Reinforcement learning (RL) is defined on a discrete-time Markov decision process (MDP) $\mathcal{M} = (\mathcal{S}, \mathcal{A}, P, r, \gamma, T)$, where $\mathcal{S}$ is the state space, $\mathcal{A}$ is the action space, $P(s'|s, a)$ is the state transition probability upon an action, $r(s, a) \in \mathbb{R}$ is the reward function, $\gamma \in (0, 1)$ is the discount factor, and $T > 0$ is the time horizon. A trajectory $\tau = (s_0, a_0, \ldots, s_{T-1}, a_{T-1})$ is a sequence of state-action pairs, and its cumulative reward is denoted by $r(\tau) = \sum_{t=0}^{T-1} \gamma^t r(s_t, a_t)$. A policy $\pi(a|s)$ defines probabilistic action choices in a given state. Denote $\pi(\tau) = \prod_{t=0}^{T-1} P(s_{t+1}|s_t, a_t)\pi(a_t|s_t)$ the probability of a trajectory under the policy $\pi$. Forward RL seeks to find an optimal policy that maximises the expected return $\mathcal{J}_r(\pi) = \mathbb{E}_{\tau \sim \pi}[r(\tau)]$, but there might be more than one optimal policy. Maximum Causal Entropy (MaxEnt) RL solves this ambiguity by augmenting the expected return with a *causal entropy* (Ziebart et al., 2010) term $\mathcal{H}(\pi) = \mathbb{E}_{\tau \sim \pi}[\sum_{t=0}^{T-1} -\gamma^t \log \pi(a_t|s_t)]$, i.e., the

objective is to find a policy to maximise $\mathcal{J}_r(\pi) + \beta \mathcal{H}(\pi)$, which is more likely to be unique. Here, $\beta > 0$ controls the relative importance of reward and entropy and, without loss of generality, it is often assumed $\beta = 1$ (Yu et al., 2019b).

Suppose the reward function is unknown, but we have a set of demonstrated trajectories sampled from an *unknown* expert policy $\pi_E$. Inverse RL (IRL) aims to infer a reward function such that when integrated with $\mathcal{M} \setminus r$ ($\mathcal{M}$ without $r$), the optimal policy induced by the learned reward function will generate the same behaviour as demonstrations. In MaxEnt RL, the probability of a trajectory $p(\tau)$ will follow a power law w.r.t. $r(\tau)$, i.e., the probability of a trajectory $p(\tau)$ is proportional to the exponential of $r(\tau)$ (Haarnoja et al., 2017), MaxEnt IRL (Ziebart et al., 2008; 2010) thus reduces to the following maximum likelihood estimation problem for a parameterised reward function $r_\theta$:

$$p_\theta(\tau) = \frac{1}{\mathcal{Z}_\theta} \exp(r_\theta(\tau)), \quad \max_\theta \mathbb{E}_{\tau \sim \pi_E}[\log p_\theta(\tau)] = \mathcal{J}_\theta(\pi_E) - \log \mathcal{Z}_\theta. \quad (1)$$

Here, $\mathcal{Z}_\theta = \int \exp(r_\theta(\tau))d\tau$ is the *partition function* of the trajectory distribution $p_\theta(\tau)$, which is intractable to compute for large or continuous states-action spaces.

Finn et al. (2016a); Fu et al. (2018) introduced Adversarial IRL (AIRL) as a sampling-based approximation to MaxEnt IRL. AIRL reframes the MaxEnt IRL problem using a generative adversarial network setup (Goodfellow et al., 2014). It uses a discriminator $D_\theta$ (a binary classifier) and a parameterised policy $\pi_\omega$ as an adaptive sampler. The discriminator is in a particular form:

$$D_\theta(s, a) = \exp(f_\theta(s, a))/(\exp(f_\theta(s, a)) + \pi_\omega(a|s)), \quad (2)$$

where $f_\theta$ will serve as the learned reward function. The update of $D_\theta$ is interleaved with the update of $\pi_\omega$: $D_\theta$ is trained to update the reward function by distinguishing between the trajectories sampled from the expert and the adaptive sampler:

$$\max_\theta \mathbb{E}_{\tau \sim \pi_E}[\log D_\theta(s, a)] + \mathbb{E}_{\tau \sim \pi_\omega}[1 - \log D_\theta(s, a)], \quad (3)$$

while $\pi_\omega$ is updated to maximise rewards with entropy: $\max_\omega \mathcal{J}_\theta(\pi_\omega) + \mathcal{H}(\pi_\omega)$. At optimality, $D_\theta(s, a)$ will converge to 0.5 for all $(s, a)$ pairs and $f_\theta$ will recover the true reward up to a constant, under certain conditions (Fu et al., 2018).

## 3.2 Inverse Constrained Reinforcement Learning

Constrained RL (CRL) operates on a constrained MDP (CMDP) $\mathcal{M}^c = (\mathcal{S}, \mathcal{A}, P, r, \gamma, T, c, \alpha)$ that extends an MDP $\mathcal{M}$ with a cost function $c : \mathcal{S} \times \mathcal{A} \to \mathbb{R}$ and a budget $\alpha \geq 0$ (Altman, 2021). The goal is to find a policy that maximises rewards $\mathcal{J}_r(\pi)$ while ensuring the expected costs $\mathcal{J}_c(\pi) = \mathbb{E}_{\tau \sim \pi}[c(\tau)] \leq \alpha$, where $c(\tau) = \sum_{t=0}^{T-1} c(s_t, a_t)$. When $\alpha = 0$, hard constraints apply and all state-action pairs with non-zero (positive) costs are strickly prohibited to visit. The cost function thus reduces to a binary function: $c(s, a) \in \{0, 1\}$, where 1 indicates any positive costs. Hard constraints simplify optimisation while capturing safety or other broad constraints imposed by physical laws. For $\alpha > 0$, soft constraints allow limited violations within the budget $\alpha$. Inverse CRL (ICRL) aims to recover the cost function from expert demonstrations such that the optimal policy under $\mathcal{M}^c \setminus c$ reproduces the demonstrations, typically assuming a *known* reward $r$. ICRL uses the MaxEnt principle and focuses on hard constraints, introducing a *feasibility function* $\delta_\phi(s, a) \in \{0, 1\}$ (1 for feasible and 0 for not) and reducing to the following problem:

$$p_\phi(\tau) = \frac{1}{\mathcal{Z}_\phi} \exp(r(\tau)) \cdot \delta_\phi(\tau), \quad \max_\phi \mathbb{E}_{\tau \sim \pi_E}[\log p_\phi(\tau)] = \max_\phi [\log \delta_\phi(\tau)] - \log \mathcal{Z}_\phi. \quad (4)$$

Here, $\delta_\phi(\tau) = \prod_{t=0}^{T-1} \delta_\phi(s_t, a_t)$ denotes the feasibility of a trajectory, i.e., $\tau$ is feasible if and only if $\delta_\phi(s_t, a_t) = 1, \forall t < T$. The partition function $\mathcal{Z}_\phi = \int \exp(r(\tau)) \cdot \delta_\phi(\tau)d\tau$ depends only on the feasibility function parameter $\phi$. A cost function can be obtained from $\delta_\phi(s, a)$ by $c(s, a) = 1 - \delta_\phi(s, a)$. For compatibility with gradient-based optimisation, in practice, ICRL often models $\delta_\phi(s, a)$ as a continuous function in the range $(0, 1)$ and uses a very small $\alpha$, i.e., it solves a soft version of the problem as an approximation to the hard version (Malik et al., 2021).

## 4 Problem Formulation

Our goal is to integrate IRL and ICRL into a unified framework capable of inferring *both unknown* rewards and constraints from demonstrations. We start by formalising the problem on CMDP $\mathcal{M}^c$

with hard constraints, following the ICRL convention. Suppose neither the reward nor cost function is known, but we have a set of demonstrations $\mathcal{D}_E = \{\tau\}$ sampled from an expert policy $\pi_E^c$. We aim to infer *both* a reward function and a cost function, such that when integrated with $\mathcal{M}^c \setminus \{r, c\}$, the optimal policy reproduces the demonstrations. To achieve this, we introduce two key desiderata:

**Desideratum 1.** Simultaneous and efficient inference of rewards and constraints.

**Desideratum 2.** Prior knowledge of the correlations between rewards and constraints.

The first desideratum ensures efficient simultaneous inference of both rewards and constraints, while the second incorporates prior knowledge of their relationship, embedding it into the inference process to align with observed behaviours. Without this, inferred functions may deviate from the ground truth. In Section 5, we address the first desideratum by developing an adversarial learning framework for joint reward-constraint inference, and in Section 6, we embed insights from animal behaviour to inform the reward-constraint correlations.

## 5 Adversarially Simultaneous Reward-Constraint Inference

### 5.1 Extending MaxEnt IRL with Trajectory-level Feasibility Functions

We build our model on CMDP while assuming both the reward and cost functions are unknown. Following the ICRL convention, we consider hard constraints and use a binary feasibility function to represent constraints. Previous ICRL methods define a feasibility function, $\delta(s, a)$, on state-action pairs (Malik et al., 2021; Xu & Liu, 2023). However, intuitively, states/actions within a trajectory can be causally related; one state/action may depend on or influence others, suggesting that constraints may exist not only for individual states/actions but also among them. This implies that the overall feasibility of a trajectory, $\delta(\tau)$, is not simply the product of independent $\delta(s_t, a_t)$ evaluations, but rather a more complex function that incorporates the sequence of those over time. To capture this complexity, we use a feasibility function $\bar{\delta}(\tau)$ that operates directly at the trajectory level, providing greater expressive power. We experimentally examine this trajectory-centre definition in Section 7.

Under the MaxEnt IRL framework with $r_\theta(s, a)$ and $\bar{\delta}_\phi(\tau) \in \{0, 1\}$ being the parameterised reward and trajectory-level feasibility functions, the generation process of trajectories induced by an optimal policy can be characterised by the following distribution:

$$p_{\theta,\phi}(\tau) = \frac{1}{\mathcal{Z}_{\theta,\phi}} \exp(r_\theta(\tau)) \cdot \bar{\delta}_\phi(\tau), \quad \mathcal{Z}_{\theta,\phi} = \int \exp(r_\theta(\tau)) \cdot \bar{\delta}_\phi(\tau) \, d\tau. \tag{5}$$

Given a set of expert demonstrations $\mathcal{D}_E$, the problem of inferring $r_\theta$ and $\bar{\delta}_\phi$ can be cast as a maximum likelihood estimate problem w.r.t. the probability distribution $p_{\theta,\phi}(\tau)$ defined above:

$$\max_{\theta,\phi} \mathbb{E}_{\tau \sim \mathcal{D}_E} [\log(p_{\theta,\phi}(\tau))] = \mathbb{E}_{\tau \sim \mathcal{D}_E} [r_\theta(\tau) + \log \bar{\delta}_\phi(\tau)] - \log \mathcal{Z}_{\theta,\phi}. \tag{6}$$

### 5.2 Building the Adversarial Learning Framework

For efficiency, we want to adopt the idea of AIRL to recast the problem in Eq. (6) as optimising a generative adversarial network and solve it by interchangeably updating a discriminator $D_{\theta,\phi}$ and a sampler $q_{\omega,\phi}$ until $D_{\theta,\phi}$ cannot distinguish between demonstrations and the behaviour generated by $q_{\omega,\phi}$. Since we adopt a trajectory-level feasibility function, $D_{\theta,\phi}$ should also be trajectory-centric and we take a particular form for it, resembling its counterpart in AIRL:

$$D_{\theta,\phi}(\tau) = \left( \exp(f_\theta(\tau)) \cdot \bar{\delta}_\phi(\tau) \right) / \left( \exp(f_\theta(\tau)) \cdot \bar{\delta}_\phi(\tau) + q_{\omega,\phi}(\tau) \right). \tag{7}$$

The discriminator is trained to maximise the following objective to distinguish between the demonstrated and generated trajectories:

$$\mathbb{E}_{\tau \sim \pi_E^c} [\log D_{\theta,\phi}(\tau)] + \mathbb{E}_{\tau \sim q_{\omega,\phi}} [\log(1 - D_{\theta,\phi}(\tau))]. \tag{8}$$

In principle, the sampler $q_{\omega,\phi}$ can be any function approximator, such as a Gaussian process or a neural network, that can predict the probability of a trajectory. Specifically, we use a form for $q_{\omega,\phi}$ that factorises the joint effect of both the policy and constraints in trajectory generation:

$$q_{\omega,\phi}(\tau) = \pi_\omega(\tau) \cdot \bar{\delta}_\phi(\tau). \tag{9}$$

It generates a trajectory using the policy $\pi_\omega$ only if $\bar{\delta}_\phi(\tau) = 1$, ensuring that the trajectory meets the constraints, and only valid, constraint-compliant trajectories are sampled during training. This form offers two key advantages: First, it simplifies optimisation by allowing the use of existing forward RL methods to train $\pi_\omega$ to maximise entropy-regularised rewards: $\max_\omega \mathcal{J}_\theta(\pi_\omega) + \mathcal{H}(\pi_\omega)$. Second, since $\pi_\omega$ and $\bar{\delta}_\phi$ are independent, $\bar{\delta}_\phi$ cancels out in the discriminator $D_{\theta,\phi}$, leaving it to dependent solely on the reward: $D_\theta(\tau) = \exp(f_\theta(\tau))/(\exp(f_\theta(\tau)) + \pi_\omega(\tau))$, reverting to its AIRL form, which simplifies the calculation of gradients we will introduce below.

At optimality, $D_\theta(s, a)$ will converge to $0.5$ for all $(s, a)$ pairs, indicating that $f_\theta$ will effectively function as the reward. As a result, the trajectory sampler $q_{\omega,\phi}(\tau) = \exp(f_\theta(\tau)) \cdot \bar{\delta}_\phi(\tau)$ will approach the optimal trajectory distribution $p_{\theta,\phi}(\tau)$ in Eq. (5). This, in turn, leads to solving the target problem in Eq. (6), achieving maximum likelihood estimation for reward-constraint inference.

### 5.3 SIMULTANEOUS REWARD-CONSTRAINT INFERENCE

To enable gradient-based updates for both the reward and feasibility functions, we follow the ICRL approach by approximating hard constraints with a continuous feasibility function $\bar{\delta}(\tau) \in (0, 1)$, and imposing a small budget $\alpha$ as the constraint for the discriminator objective in Eq. (8). Incorporating this constraint into the objective results in the following Lagrangian:

$$\mathcal{L}_{\text{dis}}(\theta, \phi, \lambda) = \mathbb{E}_{\tau \sim \pi_E^c}[\log D_\theta(\tau)] + \mathbb{E}_{\tau \sim q_{\omega,\phi}}[\log(1 - D_\theta(\tau))] - \lambda(\mathbb{E}_{\tau \sim \pi_\omega}[\bar{\delta}_\phi(\tau)] - \alpha). \quad (10)$$

Instead of iteratively updating the Lagrange multiplier $\lambda$ using primal-dual methods, we fix $\lambda = 1$ to take the budget term as a penalty, improving efficiency and simplifying $\mathcal{L}_{\text{dis}}(\theta, \phi, \lambda)$ to $\mathcal{L}_{\text{dis}}(\theta, \phi)$.

Taking the gradient of the final discriminator's objective $\mathcal{L}_{\text{dis}}(\theta, \phi)$ w.r.t. the reward parameter $\theta$ and estimate it on expert demonstrations $\mathcal{D}_E$ and generated trajectories $\mathcal{D}_S$ gives $\frac{\partial}{\partial \theta} \mathcal{L}_{\text{dis}}(\theta, \phi) =$

$$\mathbb{E}_{\tau \sim \mathcal{D}_E} \left[ \left( 1 - \frac{\exp(f_\theta(\tau))}{\exp(f_\theta(\tau)) + \pi_\omega(\tau)} \right) \frac{\partial f_\theta(\tau)}{\partial \theta} \right] - \mathbb{E}_{\tau \sim \mathcal{D}_S} \left[ \frac{\exp(f_\theta(\tau))}{\exp(f_\theta(\tau)) + \pi_\omega(\tau)} \frac{\partial f_\theta(\tau)}{\partial \theta} \right]. \quad (11)$$

Similarly, the gradient of $\mathcal{L}_{\text{dis}}(\theta, \phi)$ w.r.t. the feasibility parameter $\phi$ can be estimated as

$$\frac{\partial}{\partial \phi} \mathcal{L}_{\text{dis}}(\theta, \phi) = \mathbb{E}_{\tau \sim \mathcal{D}_S} \left[ \frac{\pi_\omega(\tau)}{1 + \exp(f_\theta(\tau))/\pi_\omega(\tau)} \frac{\partial \bar{\delta}_\phi(\tau)}{\partial \phi} \right] - \mathbb{E}_{\tau \sim \mathcal{D}_P} \left[ \frac{\partial \bar{\delta}_\phi(\tau)}{\partial \phi} \right], \quad (12)$$

where $\mathcal{D}_P$ is a set of trajectories sampled using the policy $\pi_\omega(\tau)$. The detailed derivations of Eq. (11) and Eq. (12) are given in Appendix A.

## 6 REWARD-FEASIBILITY CONSTRAST PRIOR DRAWN FROM ANIMAL BEHAVIOUR

While the adversarial learning framework facilitates the simultaneous and efficient inference of rewards and constraints, it lacks the ability to incorporate prior knowledge about the correlation between the two. This limitation can lead to inferred rewards and constraints deviating from the ground truth. To address this, we seek insights from animal behaviour, particularly meerkats, due to their complex social structures (Drewe et al., 2009; Madden et al., 2009; 2011), which offer a natural context for understanding the reward-constraint relationship. By analysing footage of a meerkat mob at a zoo (Rogers et al., 2023), we extract spatio-temporal transitions between different behaviours. The overview of our data analysis is shown in Figure 2, with detailed procedures in Appendix B.

Our analysis reveals an interesting phenomenon: most behaviour transitions occur over short distances, with transition frequencies decreasing as distance grows. However, an inflexion point emerges at certain high distances, where transitions are more frequent than expected. This suggests that long-distance movements, typically associated with low feasibility (high constraints), may occur due to strong incentives (high rewards). We hypothesise that these movements are driven by high rewards, which outweigh the constraints. This phenomenon, termed the *"reward-feasibility contrast prior,"* reflects the common understanding that higher returns often come with greater risks.

Based on this hypothesised prior, we design a regularisation term for the discriminator objective:

$$\mathcal{C}(\theta, \phi) = \varphi \cdot \mathbb{E}_{\tau \sim \mathcal{D}_E \cup \mathcal{D}_S} \left[ (f_\theta(\tau) - \bar{\delta}_\phi(\tau))^2 \right], \quad (13)$$

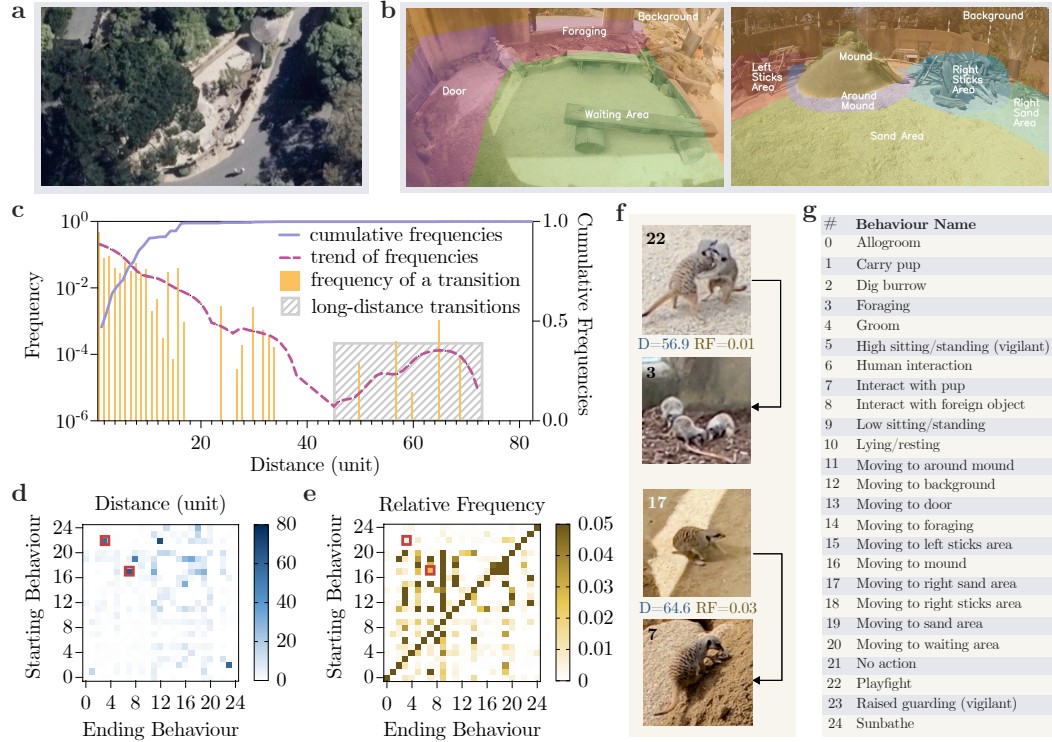

Figure 2: Illustrations, statistics and analysis of the meerkat behaviours. **(a)** An aerial view of the meerkat's habitat. **(b)** The habitat is divided into various areas shaded with different colours. **(c)** The histogram of behaviour transition distances with its trend line and cumulative values. An inflexion point is observed at a high distance, suggesting that long-distance transitions are likely driven by higher rewards. **(d)** The average distance a meerkat moves during a behaviour transition. **(e)** The relative frequency of transitions by the starting behaviour, highlighting two long-distance transitions beyond the inflexion point with bold boxes. **(f)** Detailed illustrations with distance and relative frequency values for two examples of long-distance transitions. **(g)** Numerical encoding for each behaviour. Note that "moving to an area" is treated as a distinct behaviour because movement does not always lead to reaching the intended destination. The meerkat may change course midway, stop, or transition into a different behaviour before arriving.

where $\varphi > 0$ is a constant coefficient. It encourages rewards and feasibilities to be polarised, enhancing their disparity. To enable gradient-based optimisation, it is framed as an expectation. Note that this prior is independent of our proposed framework and can be embedded into existing inverse constraint inference methods, such as ICRL, to enhance their performance, as evidenced by our experimental results in the next section. In addition, drawing on (Malik et al., 2021), we incorporate the following regularisation term to encourage the search for the minimum constraint that can match the expert trajectory while avoiding overfitting during training: $\kappa \cdot \mathbb{E}_{\tau \sim \mathcal{D}_S} |1 - \bar{\delta}_\phi(\tau)|$, where $\kappa > 0$ is the coefficient. As a summary, the entire training process is presented in Algorithm 1.

## 7 EXPERIMENTS

We aim to address two key questions through our experiments: (1) Can AIRCL accurately infer reward functions and constraints from observed behaviour? (2) Can the reward-feasibility contrast prior be generalised to other environments beyond animal behaviour modelling, and can it be integrated into other algorithms to enhance their performance?

We evaluate AIRCL across various constrained environments, including a grid world with random obstacles, three simulated robot control tasks with safety constraints using the MuJoCo physics

---

**Algorithm 1** Adversarial Inverse Reward-Constraint Learning

---

1: **Input:** Expert trajectories $\mathcal{D}_E = \{\tau\}$. Initial parameters of $f_\theta(\tau)$, $\bar{\delta}_\phi(\tau)$ and $\pi_\omega(\tau)$.
2: **repeat**
3:  Generate set of sampled trajectories $\mathcal{D}_S$ using the sampler $q_{\omega,\phi}(\tau) = \pi_\omega(\tau) \cdot \bar{\delta}_\phi(\tau)$.
4:  Generate set of sampled trajectories $\mathcal{D}_P$ using the policy $\pi_\omega(\tau)$.
5:  Update $\theta$ to increase $\mathcal{L}_{\mathrm{dis}}(\theta, \phi)$ with gradients $\frac{\partial}{\partial\theta}\mathcal{L}_{\mathrm{dis}}(\theta, \phi)$ estimated on $\mathcal{D}_E$ and $\mathcal{D}_S$.
6:  Update $\phi$ to increase $\mathcal{L}_{\mathrm{dis}}(\theta, \phi)$ with gradients $\frac{\partial}{\partial\phi}\mathcal{L}_{\mathrm{dis}}(\theta, \phi)$ estimated on $\mathcal{D}_S$ and $\mathcal{D}_P$.
7:  Update $\omega$ by using the forward RL methods with the reward function $r_\theta(\tau)$.
8: **until** Convergence or manual termination
9: **Output:** Reward function $f_\theta(\tau)$, feasibility function $\bar{\delta}_\phi(\tau)$ and policy $\pi_\omega(\tau)$.

---

engine (Liu et al., 2022), and real-world social animal behaviour modelling (Rogers et al., 2023), which exhibit spatio-temporal causal constraints (Gendron et al., 2023). We compare our method against the following state-of-the-art imitation learning, IRL and ICRL methods:

- **Generative Adversarial Constraint Learning (GACL)** (Liu et al., 2022) follows the design of Generative Adversarial Imitation Learning (Ho & Ermon, 2016), training a policy that mirrors expert behaviour while inferring constraints using a modified reward model, $r'(s, a) = r(s, a) + \log \delta(s, a)$, where infeasible actions/states are penalised with $-\infty$.

- **Adversarial Inverse Reinforcement Learning (AIRL)** (Fu et al., 2018) uses an adversarial process to learn robust reward functions against changes in environment dynamics.

- **(Variational) Inverse Constrained Reinforcement Learning (ICRL\*)** is our implementation, combining the efficient optimisation of vanilla ICRL (Malik et al., 2021) with the ability of Variational ICRL (Liu et al., 2022) to infer the distribution of feasibility $p(\delta|s, a)$, capturing the epistemic uncertainty. We achieve this by removing the small budget constraint $\alpha$ in ICRL and inferring a continuous feasibility function $\delta(s, a) \in (0, 1)$, allowing for a more flexible feasibility distribution. Note that ICRL\* requires known reward functions.

We also introduce variations of ICRL\* and AIRCL to assess the generality of the reward-constraint contrast prior, the expressiveness of trajectory-level feasibility, and the advantages over linear assumption methods (Lindner et al., 2024): ICRL with the contrast prior (**ICRL$_{+\mathbf{prior}}$**), AIRCL without the contrast prior (**AIRCL$_{-\mathbf{prior}}$**), AIRCL with linear reward and constraint functions (**AIRCL$_{\mathbf{linear}}$**), and AIRCL with the state-action-level feasibility function (**AIRCL$_{\mathbf{sa}}$**). We use proximal policy optimisation (PPO) (Schulman et al., 2017) for policy optimisation in all methods. Key hyperparameters and architecture details are provided in Appendix D.

## 7.1 GRID WORLD

To preliminarily validate AIRCL, we test it in a 6×6 grid world environment to evaluate its ability to recover the reward and feasibility functions. The environment, shown in Figure 3, includes five obstacles, and the agent's goal is to reach the destination from the start without hitting obstacles while minimising distance. Following (Xu & Liu, 2023), we add a 1% chance of random actions to introduce stochasticity into the agent's behaviour.

We evaluate the algorithm's effectiveness in recovering rewards and constraints using the following two metrics: (1) **Reward-Constraint Accuracy (RCAcc):** Following (Gleave et al., 2022), we evaluate the accuracy of recovered rewards and constraints by the discriminator's ability to differentiate expert and generated trajectories. A value closer to 1 indicates that the restored reward function, feasibility function, and policy enhance the discriminator's judgment accuracy. It is calculated by $\mathrm{RcAcc} = \mathbb{E}_{\tau \sim \mathcal{D}_E}[\mathbf{1}_{\tau\text{ predicted to be true}}]$. (2) **Constraint Violation Rate (VioRate):** It measures the proportion of timestamps where predefined constraints are violated in generated trajectories. A lower rate indicates better adherence to constraints. Note that our AIRCL uses the sampler $q_{\omega,\phi}(\tau) = \pi_\omega(\tau)\bar{\delta}_\phi(\tau)$ to generate trajectories. We calculate this constraint violation rate by $\mathrm{ViolRate} = \frac{1}{NT}\sum_{i=1}^{N}\sum_{t=0}^{T-1}\mathbf{1}_{\text{violation at }(s_t, a_t)\text{ of }\tau_i}$.

After 10 independent trials, we measure the discriminator accuracy and constraint violation rate for AIRCL and baseline methods. Table 1 presents the average and standard deviation for these

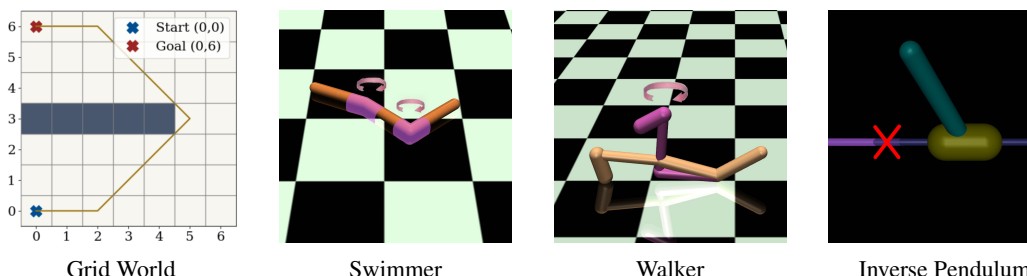

Grid World · Swimmer · Walker · Inverse Pendulum

Figure 3: In the grid world, dark grids are obstacles, and the yellow line shows the optimal trajectory. In MuJoCo virtual robot control tasks, Swimmer and Walker have restricted torque, and Inverse Pendulum has prohibited positions. Constraints are unknown to agents before training.

Table 1: Discriminator accuracy and constraint violation rates across grid world and robot tasks.

| Method | Reward-Constraint Accuracy | | | | Constraint Violation Rate | | | |
|---|---|---|---|---|---|---|---|---|
| | Grid World | Swimmer | Walker | Pendulum | Grid World | Swimmer | Walker | Pendulum |
| AIRL | 0.625±0.078 | 0.500±0.100 | 0.501±0.001 | 0.553±0.054 | 0.095±0.026 | 0.614±0.035 | **0.936**±0.006 | 0.321±0.281 |
| GACL | 0.828±0.056 | 0.635±0.104 | **0.755**±0.043 | 0.565±0.128 | 0.073±0.026 | 0.623±0.007 | 0.942±0.002 | 0.432±0.151 |
| ICRL* | – | – | – | – | 0.055±0.046 | 0.626±0.028 | 0.940±0.001 | 0.405±0.213 |
| ICRL*$_{+prior}$ | – | – | – | – | **0.051**±0.039 | 0.605±0.013 | 0.940±0.003 | 0.400±0.221 |
| AIRCL | **0.829**±0.026 | **0.769**±0.131 | 0.506±0.011 | **0.851**±0.131 | 0.071±0.036 | **0.589**±0.030 | **0.936**±0.005 | **0.214**±0.244 |
| AIRCL$_{-prior}$ | 0.635±0.201 | 0.744±0.100 | 0.510±0.013 | 0.635±0.201 | 0.081±0.036 | **0.589**±0.027 | 0.938±0.006 | 0.234±0.222 |
| AIRCL$_{linear}$ | 0.805±0.019 | 0.768±0.109 | 0.512±0.009 | 0.806±0.169 | 0.079±0.020 | 0.621±0.006 | **0.936**±0.003 | **0.214**±0.132 |
| AIRCL$_{sa}$ | 0.820±0.027 | 0.753±0.082 | 0.508±0.008 | 0.658±0.195 | 0.082±0.042 | 0.627±0.030 | 0.939±0.004 | 0.348±0.218 |

metrics. AIRCL outperforms other methods except for two ICRL* variants, with GACL showing slightly inferior results, answering the question (1). The superior performance of ICRL* is expected due to its access to ground-truth rewards. AIRCL significantly outperforms AIRL in both metrics, demonstrating strong constraint recovery abilities in discrete environments. Removing the prior led to declines in both metrics, highlighting its importance in AIRCL and thus answering the question (2).

## 7.2 ROBOT CONTROL

We test AIRCL and baseline algorithms in the MuJoCo environment using three robotic control tasks: Swimmer, Walker, and Inverse Pendulum, which are designed in (Liu et al., 2022) as benchmarks for ICRL. Safety constraints, such as torque and position constraints, are introduced, as shown in Figure 3. To simulate real-world uncertainties, random noise is added, and each experiment is repeated ten times with different random seeds, with results summarised in Table 1.

We use the same evaluation metrics as in grid worlds. In general, all methods demonstrate decreased performance on robot tasks compared to grid worlds, due to the more complex dynamics in robotic environments. For question (1), AIRCL outperforms the baselines in most tasks, particularly in the Inverse Pendulum, showing strong constraint adherence and generalisation capabilities compared to GACL. However, in the Walker task, the complex dynamics result in lower performance across all algorithms, though AIRCL still demonstrates competitive constraint adherence. For question (2), the reward-feasibility contrast prior proves effective, as ICRL* with the prior outperforms the version without it. Additionally, AIRCL's non-linear reward functions and trajectory-level feasibility offer clear advantages over simpler, linear models, especially in more complex environments. This underscores the benefits of trajectory-level feasibility and non-linear reward functions. In the Walker task, the model without the prior performs better, highlighting the limitation that the prior may not be applicable in every setting, especially in environments where such polarisation between rewards and constraints does not align with the task dynamics.

## 7.3 MEERKAT BEHAVIOUR MODELLING

Our study aims to model the social behaviours of meerkats by exploring the causal relationships driving their actions in a controlled environment. The meerkat behaviour recognition dataset (Rogers et al., 2023), comprising twenty 12-minute annotated videos, provides a comprehensive view of the

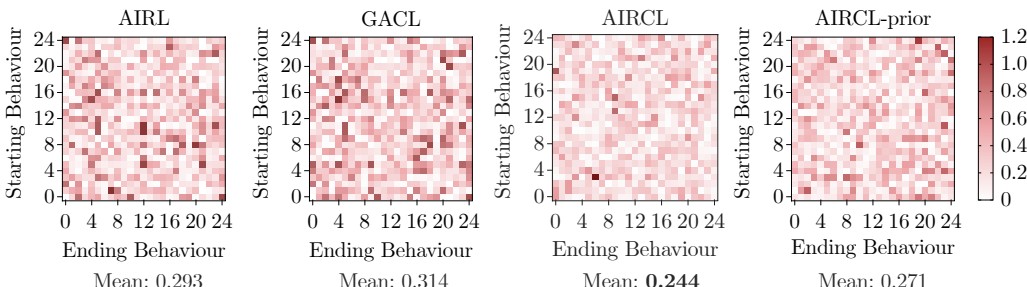

Figure 4: The causal constraint discrepancies between trajectories generated using three generative adversarial learning algorithms: AIRL, GACL, AIRCL, and real trajectories. These discrepancies are quantified using the PCMCI algorithm and represented by heatmaps. Lighter colours indicate smaller discrepancies in state transitions. The horizontal and vertical axes represent the codes for 25 initial states and 25 destination states, which can be looked up in Figure 2.

meerkats' actions at each timestep. A major challenge in behaviour modelling is identifying the true constraints that govern these actions, especially when the behavioural dynamics are influenced by hidden causal factors. To address this, we utilise the PCMCI algorithm (Runge et al., 2019) to detect causal relationships in time-series data. The strength of these relationships is visualised through heatmaps, which help us assess the actual behavioural constraints imposed on the meerkats' actions. By generating behavioural trajectories using three different generative adversarial algorithms (GACL, AIRL, and AIRCL), we employ PCMCI to analyse the causal structures within these trajectories. These generated causal heatmaps are then compared to those derived from real meerkat data. Detailed justifications for using PCMCI and the original causal constraint heatmaps are given in Appendix C. We exclude ICRL* variants from our analysis due to their reliance on ground-truth rewards, which are unavailable in real-world animal behaviour scenarios, making them unsuitable for this context.

The resulting heatmaps, presented in Figure 4, reveal the similarity between the generated causal constraints and those observed in the actual data. A comparison of the absolute differences between the generated and real causal constraints provides a clear picture of each algorithm's performance. AIRCL consistently demonstrates smaller deviations from the ground-truth constraints, indicating its effectiveness in capturing the underlying causal relationships more accurately than both GACL and AIRL. Although minor discrepancies are observed in certain state transitions, AIRCL still achieves superior constraint recovery overall, particularly in modelling complex behavioural patterns. The ablation study, where we remove the reward-constraint contrast prior, further emphasises the critical role of this prior in improving constraint recovery. The absence of the prior leads to more significant deviations, especially in key transitions, underscoring its importance for capturing subtle causal relationships in the behaviour of animals like meerkats. These findings demonstrate AIRCL's strength in recovering causal relationships in complex behavioural data, making it a promising tool for advancing the accuracy of behaviour prediction models in real-world scenarios.

## 8 CONCLUSION

In this work, we introduce AIRCL, a method for simultaneously inferring both unknown reward and constraint functions from expert demonstrations. We demonstrate AIRCL's effectiveness in both simulated robotic tasks with continuous states and actions, as well as real-world animal behaviour modelling, where it outperforms baseline methods in reward-constraint recovery and causal inference. The proposed reward-feasibility contrast prior, inspired by animal behaviour, proves critical, as its removal often leads to significant performance decline.

In a broader context, our results highlight the importance of integrating prior knowledge into inverse reward-constraint inference, as it significantly improves constraint recovery, though a prior may not be applicable in every setting. AIRCL's success across diverse tasks suggests the potential for broader applications, particularly in complex, high-dimensional environments. Future work could explore extending AIRCL to more diverse scenarios, enhancing its generalisation and scalability.

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

# A    DETAILED DERIVATION OF THE GRADIENTS OF DISCRIMINATOR'S OBJECTIVE FUNCTION

We present the detailed derivation of gradients of the discriminator objective in Eq. (10). Note that we exclude the regularisation terms introduced in Section. 6, whose gradients are straightforward to calculate. Since the reward parameter $\theta$ is only involved in the first and second terms of $\mathcal{L}_{\text{dis}}(\theta, \phi)$, its gradient w.r.t. $\theta$ is calculated by:

$$
\begin{aligned}
\frac{\partial}{\partial \theta} \mathcal{L}_{\text{dis}}(\theta, \phi) &= \frac{\partial}{\partial \theta} \mathbb{E}_{\tau \sim \pi_E^c} \left[ \log D_\theta(\tau) \right] + \frac{\partial}{\partial \theta} \mathbb{E}_{\tau \sim q_{\omega,\phi}} \left[ \log(1 - \log D_\theta(\tau)) \right] \\
&= \frac{\partial}{\partial \theta} \mathbb{E}_{\tau \sim \mathcal{D}_E} \left[ \log \frac{\exp(f_\theta(\tau))}{\exp(f_\theta(\tau)) + \pi_\omega(\tau)} \right] + \frac{\partial}{\partial \theta} \mathbb{E}_{\tau \sim \mathcal{D}_S} \left[ \log \frac{\pi_\omega(\tau)}{\exp(f_\theta(\tau)) + \pi_\omega(\tau)} \right] \\
&= \mathbb{E}_{\tau \sim \mathcal{D}_E} \left[ \frac{\partial}{\partial \theta} f_\theta(\tau) - \frac{\partial}{\partial \theta} \log(\exp(f_\theta(\tau)) + \pi_\omega(\tau)) \right] - \\
&\quad \mathbb{E}_{\tau \sim \mathcal{D}_S} \left[ \frac{\partial}{\partial \theta} \log(\exp(f_\theta(\tau)) + \pi_\omega(\tau)) \right] \\
&= \mathbb{E}_{\tau \sim \mathcal{D}_E} \left[ \left( 1 - \frac{\exp(f_\theta(\tau))}{\exp(f_\theta(\tau)) + \pi_\omega(\tau)} \right) \frac{\partial}{\partial \theta} f_\theta(\tau) \right] - \\
&\quad \mathbb{E}_{\tau \sim \mathcal{D}_S} \left[ \frac{\exp(f_\theta(\tau))}{\exp(f_\theta(\tau)) + \pi_\omega(\tau)} \frac{\partial}{\partial \theta} f_\theta(\tau) \right].
\end{aligned}
\tag{14}
$$

The feasibility function parameter $\phi$ is only involved in the second and third terms of $\mathcal{L}_{\text{dis}}(\theta, \phi)$, and thus its gradient w.r.t. $\phi$ is calculated by:

$$
\begin{aligned}
\frac{\partial}{\partial \phi} \mathcal{L}_{\text{dis}}(\theta, \phi) &= \frac{\partial}{\partial \phi} \mathbb{E}_{\tau \sim q_{\omega,\phi}} \left[ \log(1 - D_\theta(\tau)) \right] - \frac{\partial}{\partial \phi} \mathbb{E}_{\tau \sim \pi_\omega} \left[ \bar{\delta}_\phi(\tau) - \alpha \right] \\
&= \frac{1}{|\mathcal{D}_S|} \sum_{i=1}^{|\mathcal{D}_S|} \left[ \log(1 - D_\theta(\tau_i)) \frac{\partial}{\partial \phi} q_{\omega,\phi}(\tau_i) \right] - \mathbb{E}_{\tau \sim \mathcal{D}_P} \left[ \frac{\partial \bar{\delta}_\phi(\tau)}{\partial \phi} \right] \\
&= \frac{1}{|\mathcal{D}_S|} \sum_{i=1}^{|\mathcal{D}_S|} \left[ \frac{\pi_\omega(\tau_i)}{\exp(f_\theta(\tau_i)) + \pi_\omega(\tau_i)} \frac{\partial}{\partial \phi} \pi_\omega(\tau_i) \bar{\delta}_\phi(\tau_i) \right] - \mathbb{E}_{\tau \sim \mathcal{D}_P} \left[ \frac{\partial \bar{\delta}_\phi(\tau_i)}{\partial \phi} \right] \\
&= \mathbb{E}_{\tau \sim \mathcal{D}_S} \left[ \frac{\pi_\omega(\tau)}{1 + \exp(f_\theta(\tau))/\pi_\omega(\tau)} \frac{\partial}{\partial \phi} \bar{\delta}_\phi(\tau) \right] - \mathbb{E}_{\tau \sim \mathcal{D}_P} \left[ \frac{\partial}{\partial \phi} \bar{\delta}_\phi(\tau) \right].
\end{aligned}
\tag{15}
$$

# B    MEERKAT DATA PROCESSING

To obtain the meerkat behaviour, two GoPro Max cameras are set on the back wall of the enclosure, one focusing on the replica termite mound in the centre of the enclosure and the other overlooking the foraging area and entrance to the enclosure, which are hubs of activity (Figure 5). For example, the mound is a popular area for guarding behaviour, and the foraging area is popular when meerkats are looking for food. The cameras are set to automatically record videos every 12 minutes, and the contents recorded are filtered, which exclude the fragments that include visitors. Videos with many individuals, social interactions, and other interesting behaviours were selected for the annotation (Figure 6). During the annotation process, the computer vision annotation tool CVAT version 2.3 is utilised to sign the behaviour in the videos. Besides, masking techniques are used to protect the privacy of visitors and maintain the vision information of human activities at the same time. The adult and baby meerkat are annotated specifically in the dataset, with annotators using a small bounding box to note the baby meerkat's positions relative to the adults. Through multiple checks as well as using scripts to automatically detect the error, the accuracy and the consistency of the annotations are ensured (Rogers et al., 2023).

In our research, the dataset is organised according to the unique identifiers of individual meerkats, and every meerkat's behaviour is recorded over different timestamps. Specifically, the information of each

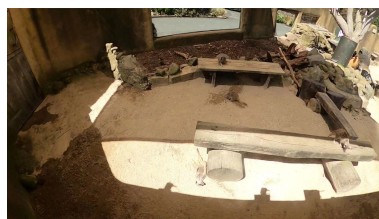 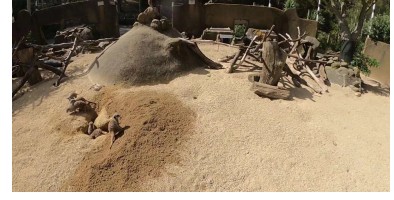

Camera view of the entrance and forag-ing area

Camera view of the mound and backside of the enclosure

Figure 5: Example images of the camera views.

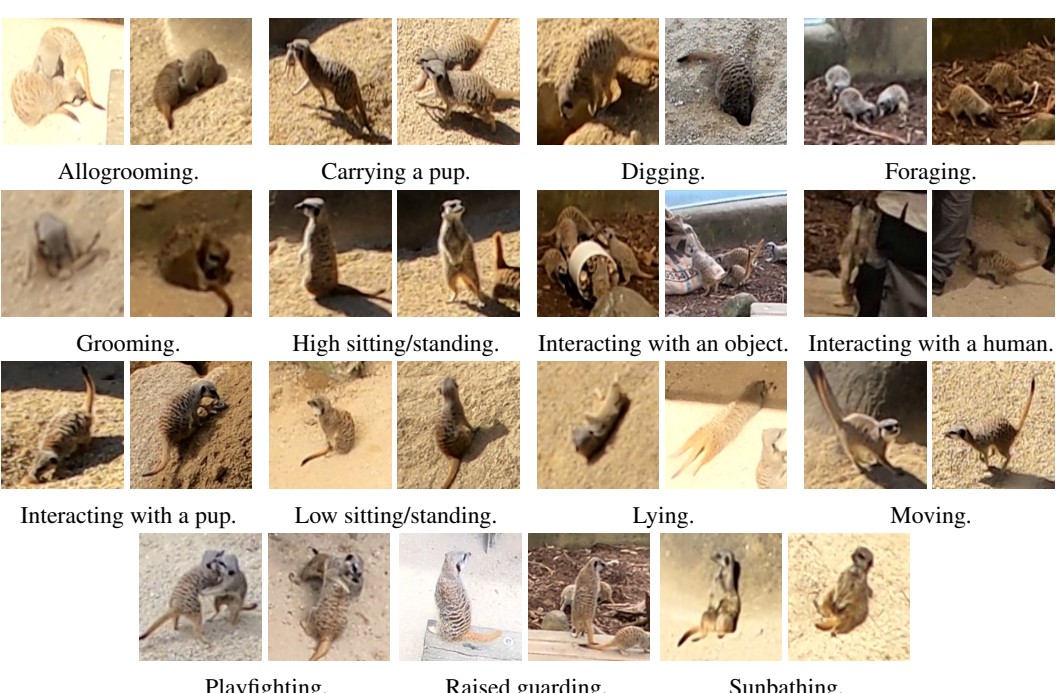

Allogrooming.      Carrying a pup.      Digging.      Foraging.

Grooming.      High sitting/standing.      Interacting with an object.      Interacting with a human.

Interacting with a pup.      Low sitting/standing.      Lying.      Moving.

Playfighting.      Raised guarding.      Sunbathing.

Figure 6: Examples of the meerkat behaviours.

timestamp includes four different parts: the identifier, the scene the meerkat is located in, the action and the three-dimensional coordinate point. In order to uniform the length of time series for analysis, we process the dataset, retaining only complete sequences of every 30 timestamps as independent trajectories, and delet those with fewer than 30 timestamps. This method can not only simplify the structure of data but also facilitate further analysis. Through this data processing approach, we construct a meerkat dataset that includes both state and action information in each timestamp.

We divide each area based on meerkat's activity range and labelled each area with a unique colour to distinguish its scope, as shown in Figure 8. After obtaining the Meerkat's behavioural dataset, we analyse the transition frequency of each area and observe that in certain areas, the activity frequency is particularly high (Figure 7). We are inspired by this to explore whether meerkat's various behaviours are driven by certain causal constraints.

## C   Causal Structure Discovery in Meerkat Behaviour

PCMCI Algorithm is designed to detect and quantify causal relationships in large-scale nonlinear time series datasets (Runge et al., 2019). Combined with the linear or non-linear conditional independence tests and causal discovery algorithm, PCMCIA can effectively improve the ability to recognise ground truth causal relationships. For example, in the meerkat behaviour dataset characterised by time series

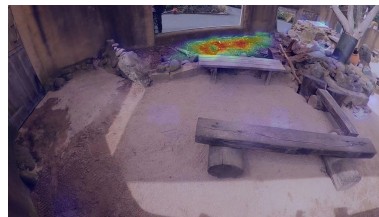 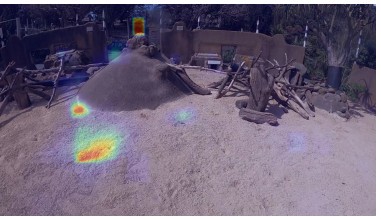

Frequency heatmap of the entrance and foraging area

Frequency heatmap of the mound and backside of the enclosure

Figure 7: The frequency of meerkat activity in various regions corresponds to the heatmap from the camera perspective. The areas where meerkat is frequently active are highlighted.

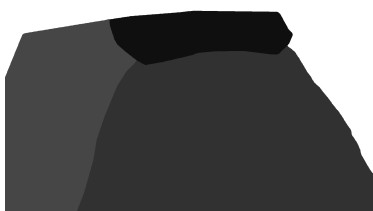 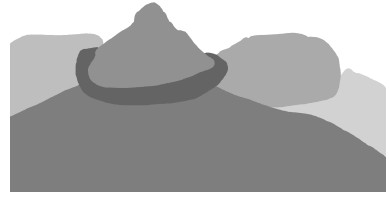

Regional division of the entrance and foraging area

Regional division of the mound and backside of the enclosure

Figure 8: Referring to Figure 2 in the main text, we have labelled blocks of different colours for each area to visually illustrate the division of meerkat activity zones.

data, PCMCI can be used to analyse the causal effects between state transitions, in case to reflect the interaction between states. In this context, behaviour transitions with higher causality might have lower constraints, while those with lower causality could show stronger constraints. This indicates that even if some state transitions offer high rewards, there may be a large cost to take the action.

In the application of PCMCI to analyse the meerkat behaviour dataset, the output is a directed graph of all states, where the colour of each edge represents the causal strength between the starting state and ending state. Considering that there are a total of 25 states, using the directed graph may cause visual confusion and make it difficult to clearly display the relationships between states. Therefore, we select heatmap to present the result of the PCMCI algorithm, and colour variations are used to display the causal strength between different states, therefore allowing a clearer display of causal differences (Figure 9).

# D    EXPERIMENT SETTINGS

We utilise the open-source library from Gleave et al. (2022), which provides high-quality, reliable, and modular implementations of various reinforcement learning and imitation learning algorithms. Built on Stable Baseline 3 (Raffin et al., 2021), the imitation library offers accurate experimental baselines, allowing us to easily train and compare a range of algorithms. We extend the library by incorporating our algorithm and modifying specific methods related to generative adversarial algorithms to support the implementation of a trajectory-based discriminator as our design.

In addition, we refer to the constrained environments and benchmarking methods designed by Liu et al. (2022) to evaluate our algorithm and baselines based on metrics of discriminator accuracy and constraint violation rate. Each constraint is customly designed to ensure that the agent performs safe and controlled actions within the defined parameters.

Furthermore, we set unique hyperparameters for each environment, optimising the algorithm's efficiency while avoiding overfitting. All important hyperparameters are listed in Table 2.

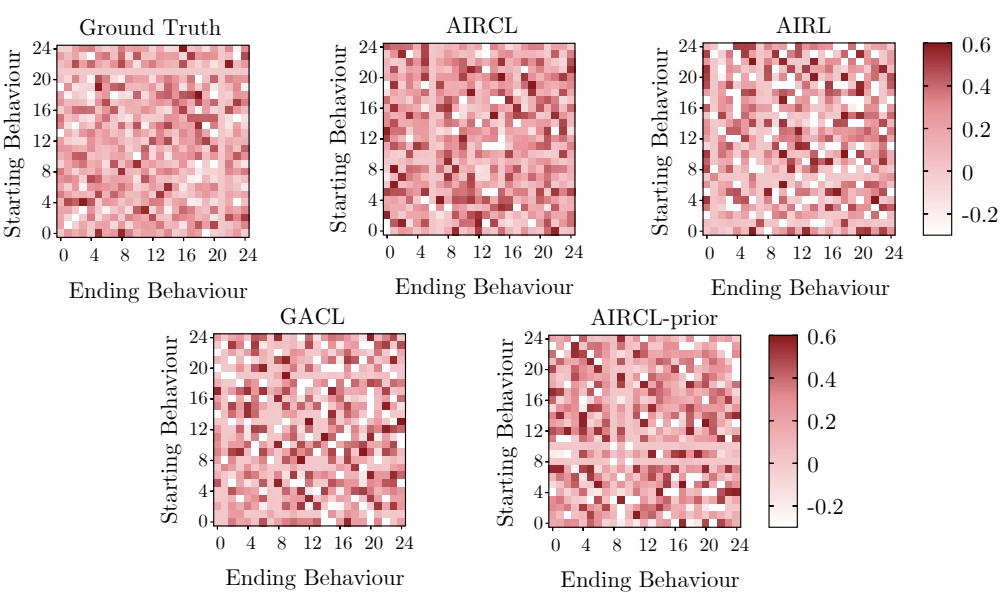

Figure 9: Causal strength for each state transition, as the ground truth constraints. Please note that in our experiments, we recorded the differences between the causal constraints of trajectories generated by each algorithm and the truth constraints.

Table 2: The hyperparameters of each environment, note that hidden units in each layer are reported for network architecture.

|  | GRIDWORLD | SWIMMER | WALKER | INVERSEPENDULUM | MEERKAT |
|---|---|---|---|---|---|
| EXPERT TRAJECTORY | 70 | 50 | 50 | 50 | 2182 |
| SAMPLED TRAJECTORY | 70 | 50 | 50 | 50 | 2182 |
| HORIZON | 10 | 500 | 500 | 100 | 30 |
| REWARD NETWORK | 32, 32 | 32, 32 | 32, 32 | 32, 32 | 32, 32 |
| FEASIBILITY NETWORK | 32, 32 | 32, 32 | 32, 32 | 32, 32 | 32, 32 |
| BATCH SIZE | 700 | 2500 | 2500 | 1000 | 500 |
| LEARNING RATE | 0.0005 | 0.0005 | 0.0005 | 0.0005 | 0.0005 |
| PPO CLIP RANGE | 0.1 | 0.1 | 0.1 | 0.1 | 0.1 |
| COEFFICIENT $(\varphi, \kappa)$ | 0.001, 0.001 | 0.001, 0.001 | 0.001, 0.001 | 0.001, 0.001 | 0.001, 0.001 |

