# OpenReview forum: "Adversarial Inverse Reward-Constraint Learning with Reward-Feasibility Contrast Prior Inspired by Animal Behaviour"
_ICLR.cc/2025/Conference — Submitted to ICLR 2025_

### Official Review · Reviewer_3fuL · 2024-10-17

**Soundness:** 2
**Presentation:** 2
**Contribution:** 2
**Rating:** 3
**Confidence:** 4

**Summary:**

This paper proposes a method that can learn both reward and constraint from demonstrations. Compared to existing methods that learn both reward and constraint, this paper can solve for nonlinear reward and nonlienar constraint. The paper also uses a hypothesis, inspired by the obervation of animals, to correlate reward and constraint. Empirical evaluations are provided to validate the effectiveness of the proposed algorithm.

**Strengths:**

The strength of this paper is the hypothesis that is inspired by the obervation from real world, which is interesting.

**Weaknesses:**

1. The main weakness lies in the problem itself, i.e., learning both reward and constraint. This problem seems unnecessary. In specific, problem (6) shows that both reward $r_{\theta}$ and constraint $\bar{\delta}_{\phi}$ are learned by optimizing $E\_{\tau\sim D\_E}[r\_{\theta}(\tau)+\log \bar{\delta}\_{\phi}(\tau)]-\log Z\_{\theta,\phi}$. Why we cannot learn a single neural model $m\_{\psi}$ such that $m\_{\psi}(\tau)=r\_{\theta}(\tau)+\log \bar{\delta}\_{\phi}(\tau)$, and call $m\_{\psi}$ reward or constraint or whatever. The point here is that problem (6) can be captured by standard IRL (that only learns reward) and standard ICRL (that only learns constraint) given that the neural model $m\_{\psi}$ can capture $r\_{\theta}(\tau)+\log \bar{\delta}\_{\phi}(\tau)$ due to the universal approximation property. In that sense, I cannot get a theoretical (or even intuitive) idea of why learning both reward and constraint is necessary. Or in other words, what is the fundamental difference between learning a single $m\_{\psi}$ and learning $r\_{\theta}$ and $\log \bar{\delta}\_{\phi}$. Can the authors provide some evidence, either theoretical or intuitive, to explain this? I acknowledge that there are some papers learning both reward and constraints. However, those papers do not explain the difference between learning a single model and learning both reward and constraint either. Before solving the problem of learning both reward and constraints, the problem itself needs to be first justified.

I would like to increase my rating if the authors can solve weakness 1 because weakness 1 is a very fundamental problem to this paper.

2. To solve the Lagrangian (10), the authors directly set the dual variable $\lambda=1$ for the sake of efficiency and simplification. I agree that ignoring $\lambda$ can make the problem easier to solve. However, this can be potentially incorrect. Usually, we cannot solve a constrained optimization problem by simply penalizing the constraint in the objective function, because the resulting policy may no longer satisfy the constraint. In this case, can the authors guarantee that the learned policy (when $\lambda=1$) satisfy the constraint, i.e., $E\_{\tau\sim \pi_{\omega}}[\bar{\delta}_{\phi}(\tau)]\leq\alpha$?

3. A limitation of this paper is that it can only solve for hard constraints because $\delta(\tau)$ is binary. There are some real-world applications that align with hard constraints, e.g., collision avoidance. However, there are also many soft-constraint scenarios. When the constraints are soft, can we easily revise this paper to solve for soft constraints?

**Questions:**

Lines 237-240 says that "However, intuitively, states/actions within a trajectory can be causally related; one state/action may depend on or influence others, suggesting that constraints may exist not only for individual states/actions but also among them. This implies that the overall feasibility of a trajectory, $\delta(\tau)$, is not simply the product of independent $\delta(s_t, a_t)$ evaluations". I do not get the intuition here. I understand that "one state/action may depend on or influence others", however, why this suggests "$\delta(\tau)$ is not simply the product of independent $\delta(s_t, a_t)$ evaluations". Do the authors mean that it is possible that all $(s_t,a_t)$ in a trajectory $\tau$ are feasible but the trajectory can be infeasible or some $(s_t,a_t)$ are infeasible but the trajectory can be feasible? Can the authors provide some concrete exmaples to better explain this? Because $\delta$ is binary, it seems that product of independent $\delta(s_t, a_t)$ can already capture all the possibilities of $\delta(\tau)$. If $\delta$ is continuous between 0 and 1, the situation may be different.

---

> ### Comment · Reviewer_3fuL · 2024-11-29
>
> Thanks for the response. However, the authors do not directly answer my questions.
>
> For the weakness 1, my question is that what is the difference between learning reward and constraints and learning a single model, or more specifically "I cannot get a **theoretical (or even intuitive)** idea of why learning both reward and constraint is necessary". When I wrote the review, I suspected that the authors might use other reference to indirectly support this claim, so that I said "I acknowledge that there are some papers learning both reward and constraints. However, those papers do not explain the difference between learning a single model and learning both reward and constraint either. Before solving the problem of learning both reward and constraints, the problem itself needs to be first justified." The authors use [2] to justify learning both reward and constraints. However, [2] does not justify why learning both reward and constraints is necessary either using theory or intuition. **You cannot use an unjustified claim to support another unjustified claim**. The other way the authors use to justify is to use the empirical results in literature. However, there are two issues. First, the empirical results in literature are contradictory. In specific, [1,2] claim that learning constraints can better capture expert behaviors, however, another paper [3] use empirical results to show that there is no difference between learning reward and learning constraints. Second, the empirical results are hard to understand because the experiments are black box. Therefore, even for purely experimental research, people still expect some theory or at least intuition to explain why the proposed method works. And this is exactly what I expect. I expect the authors to directly answer the question using theory or intuition instead of empirical results or reference.
>
> For the weakness 2, the authors do not directly answer my question either. My question is "In this case, can the authors guarantee that the learned policy (when $\lambda=1$) satisfies the constraint $E_{\tau\sim\pi_{\omega}}[\bar{\delta}_{\omega}(\tau)]\leq \alpha$?" It is a "yes or no" question that expects direct answer of "yes" or "no" and justifications. The point here is that, if the answer is no, how the authors can justify the learned constraints are useful given that even your own learned policy does not satisfy your learned constraints.
>
> [3] Simplifying constraint inference with inverse reinforcement learning

---

### Official Review · Reviewer_DjSa · 2024-11-04

**Soundness:** 2
**Presentation:** 3
**Contribution:** 2
**Rating:** 6
**Confidence:** 2

**Summary:**

This paper presents the Adversarial Inverse Reward-Constraint Learning (AIRCL) framework to jointly infer rewards and constraints in an environment. The authors try to address two main gaps in inverse reinforcement learning (IRL) by simultaneously employing adversarial learning and incorporating a reward-feasibility contrast prior inspired by animal behavior. This prior states that high rewards often coincide with high constraints, which reflects real-world behavioral patterns where agents take on greater risks for greater rewards. The paper evaluates AIRCL on simulated robot tasks and a dataset of meerkat behavior, comparing its performance to existing state-of-the-art IRL and Inverse Constrained RL (ICRL) methods. Overall, the idea of incorporating adversarial IRL with the hypothesis is interesting, but the work could be improved in experimental demonstration (see Questions below for details).

**Strengths:**

1. The overall idea and algorithm structure are generally easy to follow.

2. The incorporation of adversarial IRL and the hypothesis is motivated by the two primary challenges, both of which are indeed existing in IRL's (and ICRL's) practical implementations.

3. The experimental outcomes - along with various evaluation metrics such as reward-constraint accuracy and constraint violation rate - well support the proposed framework across different environments.

**Weaknesses:**

1. The primary concern lies with the reward-feasibility contrast prior, despite it shows effectiveness in the evaluated tasks. It is derived from animal behaviors and has not been tested in more general scenarios where high reward and low constraints align, which frequently happens in certain industrial applications.

2. Additionally, while the experimental evaluation involves simulated robot tasks and a meerkat behavior dataset, the framework’s capacity to manage uncertainties - i.e., an aspect crucial to real-world applications - remains untested. Even if the prior holds, uncertainties could lead to unexpected expert demonstrations and constraint violations, which is a more realistic scenario and may challenge the method's reliability. See Question 2 for further details.

3. It would be great to clarify in theoretical formulation, e.g., a more detailed clarification of the gradients (why can we have Equation 14?). Please also see Question 3.

**Questions:**

1. How sensitive is AIRCL’s performance to the design of the reward-feasibility contrast prior? Discussing the prior's limitation and suggesting potential adaptations for broader applicability would be valuable.

2. The empirical example should add experimental results and analysis of uncertainties.

3. In Equation 15 (Appendix A), why could $\tau$ in the second term be derived to $\tau_i$ in the third formula?

---

### Official Review · Reviewer_48hC · 2024-11-04

**Soundness:** 2
**Presentation:** 2
**Contribution:** 2
**Rating:** 6
**Confidence:** 4

**Summary:**

The paper presents the Adversarial Inverse reward-constrained learning (AIRCL) framework to infer both rewards and constraints from expert demonstrations.  In this framework, rewards and constraints are updated through policy optimization, guided by expert demonstrations. A key innovation is the introduction of the “reward-feasibility contrast prior,” a hypothesis that correlates rewards with constraints. This prior is inspired by patterns observed in animal behavior, particularly meerkats, and suggests that states with high nearby rewards are more likely to be associated with stricter constraints (lower feasibility). The authors validate our AIRCL with experiments on virtual robot control tasks with safety constraints, as well as real-world animal behavior data featuring spatiotemporal causal constraints. The results demonstrate an evident alignment with expert demonstrations and a low rate of constraint violations. Furthermore, the integration of this prior into other inverse constraint inference methods leads to improved performance, underscoring its general utility.

**Strengths:**

1. Previous work in Inverse Reinforcement Learning (IRL) and Inverse Constrained Reinforcement Learning (ICRL) has primarily focused on estimating either rewards or constraints independently. In contrast, AIRCL simultaneously infers both. This dual inference approach aligns with the direction explored in [1,2,3], addressing an important topic.

2. AIRCL introduces the reward-feasibility contrast prior, a novel method for capturing the correlation between rewards and constraints. This prior is well-motivated by observations of animal behavior, providing a biologically inspired foundation for the modeling.

3. The paper conducts extensive experiments, comparing AIRCL against multiple baseline methods across various environments. These range from virtual settings like grid-world and robotic control tasks to more realistic environments, such as behavior cloning in meerkats.

In summary, the primary contribution of this paper lies in its application to the study of animal behavior, which remains an underexplored area in the ICRL literature. While the technical contribution is fair, it is less significant than the novelty in application.

[1] Liu, S. and Zhu, M. Distributed inverse constrained reinforcement learning for multi-agent systems. In Advances in Neural Information Processing Systems (NeurIPS), 2022.

[2] Shicheng Liu and Minghui Zhu. Learning multi-agent behaviors from distributed and streaming demonstra- tions. In Advances in Neural Information Processing Systems (NeurIPS), 2023a.

[3] Daehyung Park, Michael D. Noseworthy, Rohan Paul, Subhro Roy, and Nicholas Roy. Inferring task goals and constraints using bayesian nonparametric inverse reinforcement learning. In Conference on Robot Learning, CoRL, volume 100, pp. 1005–1014, 2019.

**Weaknesses:**

1. **Insufficient Comparison and Discussion.**
References [1,2,3] also infer both reward and cost functions, yet these works are not included in the current discussion. I recommend that the authors provide a more detailed comparison between their approach and the methods described in [1,2,3], highlighting key differences in methodology, performance, and applicability.
2. **Lack of Evidence for Main Contributions.**
- The authors claim that AIRCL demonstrates strong data and training efficiency; however, the paper lacks evidence to support this claim. Specifically, there is no theoretical complexity analysis or experimental metrics provided to demonstrate faster convergence rates or lower sample complexity. Addressing this gap would strengthen the paper’s argument.
- While the reward-feasibility contrast prior is presented as a central contribution of this work, its impact is not sufficiently demonstrated through the experiments. The current ablation study (with and without the prior) is too simplistic. I suggest the authors include a more thorough analysis, such as a correlation study between rewards and costs, to better validate the prior's effectiveness.
3. **Limited Applicability Due to Hard-Constraint Assumptions.**
The authors’ focus on hard-constraint scenarios may restrict the broader applicability of their method, particularly in practical settings where soft-constraint scenarios are more prevalent. Expanding the framework to handle soft constraints would enhance its relevance to a wider range of real-world problems.

**Questions:**

1. It appears that the authors may have misunderstood or misstated [1,2,3]. Specifically, [1] introduces a bi-level algorithm where learners collaboratively estimate constraints in the outer loop while learning reward functions in the inner loop. This contradicts the authors’ assertion in line 49 that "most existing ICRL methods rely on the assumption of known reward functions [1]." Consequently, the statement in the abstract that "the simultaneous inference of both remains unexplored" may be inaccurate and should be revised.

2. The animal behavior inspiration is intriguing, but I’m curious about how the behaviors in Figure 2(g) are classified. Different behaviors likely correspond to varying travel distances, so did the authors consider this aspect in the context of studying long-distance transitions?

3. The reward-feasibility contrast prior posits that higher returns are often associated with greater risks. However, this relationship may not universally apply, as trajectories with low returns can also incur high costs. I invite the authors to provide additional commentary on this point.

---

### Official Review · Reviewer_R4Xh · 2024-11-05

**Soundness:** 1
**Presentation:** 1
**Contribution:** 2
**Rating:** 3
**Confidence:** 3

**Summary:**

This paper proposes Adversarial Inverse Reward-Constraint Learning (AIRCL), a novel approach capable of simultaneously inferring both rewards and constraints. Traditional inverse reinforcement learning (IRL) and inverse constrained reinforcement learning (ICRL) methods have been limited by their prerequisite knowledge of either constraints or reward functions to infer the other. This study attempts to overcome this limitation by leveraging a reward-feasibility contrast prior to concurrently inferring both reward functions and constraints.

The paper demonstrates the efficacy of AIRCL through comparative analyses in Mujoco and Grid World environments. The proposed method outperforms other comparison methods in terms of **reward-constraint accuracy** and **constraint violation rate**. Furthermore, the study extends its application to real-world scenarios by analyzing the relationship between rewards and constraints using Meerkat data.

**Strengths:**

- The idea the authors suggested is interesting.
- The study employs a novel approach by defining both reward and feasibility functions at the trajectory level. Also, by incorporating an adversarial learning framework, the researchers enhanced the expressive power of their model.
- The empirical validation of the proposed method is conducted in Grid World and Mujoco. These environments are well-suited for simultaneous setting and experimentation of both constraints and rewards, providing a robust and controlled testing ground for the proposed algorithm.

**Weaknesses:**

The attempt to analyze meerkat data and apply insights from the analysis to this algorithm was good. However, the study fails to explain the analysis results sufficiently and logically. Please refer to the questions for detailed weaknesses.

**Questions:**

## Regarding Meerkat data
1. What are behavior transition distances? Does "unit" mean pixel-level distance when measuring distance? Figure 2 is difficult to understand. How are the distances described in Figure 2f and relative frequency values defined? Please explain more about Figures 2c, d, and e.
2. Regarding 'moving to <Place>' behaviors as described in Figure 2g, how are the distances defined? Are snapshots taken during ‘movement’ to measure distances? How can 'moving to <Place>' be either a 'starting' or 'ending' behavior?
3. Figure 2c categorizes various behaviors’ frequency only by distance, making it impossible to determine whether behaviors showing both long-distance and high frequency are the same behaviors. I expect Figures 2d and e to solve the problem, but in Figures 2d and e, while the bold boxes clearly indicate high distance, it's unclear whether they correspond to high-frequency behaviors.
4. Using PCMCI, the authors interpret that behaviors with higher causality would have low constraints (I will refer to it as A type behavior), while the behaviors with lower causality would have stronger constraints (I will refer to it as B type). However, it is hard to say that outlier behaviors (B type) have stronger constraints than routine behaviors (A type).
Also, connecting causality and constraints seems illogical since these can vary depending on external factors (human, weather). Furthermore, while the authors previously discussed that longer behavior transition distances correlate with high constraints, the sudden introduction of causal relationships seems unexplained.
5. Why were meerkats, which have complex social structures, used to study reward-constraint relationships? In my opinion, animals that prioritize instinctive behaviors over social ones would be better candidates for reward-constraint relationship studies.
6. The caption for Figure 4 appears to be incorrect. There is no heatmap about real trajectories. Also, what type of mean is being displayed in each heatmap?

## Others
1. Where in Algorithm 1 would the discriminator objective from Equation 13 be placed? And how does equation 13 polarize reward and feasibility? How is it related to 'reward-feasibility contrast prior', which claims that lower feasibility corresponds to higher reward?
2. Additionally, for the regularization term introduced in L367, please specify where it is applicable in Algorithm 1.

---

### Meta-Review · Area_Chair_wj8u · 2024-12-17

**Metareview:**

The authors propose a framework that infers reward and constraints simultaneously by using expert demos. However, insufficient comparisons are made to show the superiority of the proposed method. Moreover, the clarity of the theoretical foundations needs improvement. After reading the rebuttal, I find the authors still do not provide a good intuition for the question: why learning rewards and constraints at the same time?  We encourage the authors to polish the manuscript, motivate the idea, and resubmit to another conference.

**Additional Comments On Reviewer Discussion:**

The authors address some of the concerns of the reviewers. However, the core problems remain.

---

### Decision · Program_Chairs · 2025-01-22

Reject